# Enhancing the Efficiency of Mild-Temperature Photothermal Therapy for Cancer Assisting with Various Strategies

**DOI:** 10.3390/pharmaceutics14112279

**Published:** 2022-10-24

**Authors:** Pei Wang, Biaoqi Chen, Yunyan Zhan, Lianguo Wang, Jun Luo, Jia Xu, Lilin Zhan, Zhihua Li, Yuangang Liu, Junchao Wei

**Affiliations:** 1School of Stomatology, Nanchang University, Nanchang 330006, China; 2Jiangxi Province Key Laboratory of Oral Biomedicine, Nanchang 330006, China; 3Jiangxi Province Clinical Research Center for Oral Diseases, Nanchang 330006, China; 4Institute of Pharmaceutical Engineering, College of Chemical Engineering, Huaqiao University, Xiamen 361021, China

**Keywords:** mild-temperature photothermal therapy, immunotherapy, heat shock proteins, nanoplatforms, thermal resistance

## Abstract

Conventional photothermal therapy (PTT) irradiates the tumor tissues by elevating the temperature above 48 °C to exert thermal ablation, killing tumor cells. However, thermal ablation during PTT harmfully damages the surrounding normal tissues, post-treatment inflammatory responses, rapid metastasis due to the short-term mass release of tumor-cellular contents, or other side effects. To circumvent this limitation, mild-temperature photothermal therapy (MTPTT) was introduced to replace PTT as it exerts its activity at a therapeutic temperature of 42–45 °C. However, the significantly low therapeutic effect comes due to the thermoresistance of cancer cells as MTPTT figures out some of the side-effects issues. Herein, our current review suggested the mechanism and various strategies for improving the efficacy of MTPTT. Especially, heat shock proteins (HSPs) are molecular chaperones overexpressed in tumor cells and implicated in several cellular heat shock responses. Therefore, we introduced some methods to inhibit activity, reduce expression levels, and hinder the function of HSPs during MTPTT treatment. Moreover, other strategies also were emphasized, including nucleus damage, energy inhibition, and autophagy mediation. In addition, some therapies, like radiotherapy, chemotherapy, photodynamic therapy, and immunotherapy, exhibited a significant synergistic effect to assist MTPTT. Our current review provides a basis for further studies and a new approach for the clinical application of MTPTT.

## 1. Introduction

Hyperthermia was used to treat breast tumors in Egypt, tracing back to 5000 B.C. Tumor tissue presents increased blood vessels, blood stasis, poor heat dissipation, high resistance, difficult heat dissipation, easy heat accumulation, and rapid temperature increase [1,2]. Thus hyperthermia is effective for tumor treatment [3]. Photothermal therapy (PTT) is a kind of thermal therapy whereby light energy is converted into heat energy to improve the temperature of lesions to achieve a therapeutic effect [4,5]. Exogenous photothermal agents (PAs) are not necessary for PTT but can improve the efficiency and efficacy of therapy [6]. PTT is widely applied for the treatment of various types of tumors by promoting apoptosis or necrosis of tumor cells at high temperatures [4,7,8]. PTT relying on the introduction of an exogenous laser can achieve high accuracy, high efficiency, mild toxicity, and non-invasive treatment compared with traditional chemotherapy, radiotherapy, and surgery [5,9,10]. In addition, the laser can be used as a “light-trigger switch” to achieve remote drug control release (light stimulation response) [11,12]. In contrast, the heat can destroy the lysosome to help the drug-loaded to escape from the lysosome. Nowadays, a division between the concentration of preclinical and clinical PTT research is obvious, with preclinical studies focused on new PAs, whereas clinical studies concentrated on the exploitation of integrated laser devices [6]. The difference may reflect the fact that the effectiveness of PTT can easily be demonstrated in preclinical research, enabling the preparation and application of a wide variety of novel nanomaterials. Nevertheless, PAs hold potential in clinical transformation on account of better selectivity for the target tissue, enabling the utilization of lower-power lasers and simplifying device design. Previous studies have made significant efforts to optimize PAs by modulating the shape, size, and surface chemistry of nanoparticles [7,13,14]. Moreover, the rapid development of nanotechnology has increased advances in PTT through the development of multi-functional nanoparticles [15]. For instance, plasmonic nanoparticles, like gold nanoparticles, and platinum nanoparticles, are chosen as PAs in many reports [16,17]. In addition, synergistic therapy with PTT improves the therapeutic effect of PTT against tumors [18]. PTT directly kills tumor cells or enhances other therapies by promoting drug delivery, stimulating release, mediating tumor microenvironment (TME), eliciting tumor-specific antigen release, or modulating other biologically related responses [19,20,21,22,23,24,25].

However, the clinic application of PTT has been hindered to some extent by several limitations. For instance, it is challenging to completely kill tumor cells using PTT, thus augmenting the risk of tumor recurrence and metastasis owing to limited tissue penetration of the laser (NIR-I widow laser 1~2 cm, NIR-II widow laser > 2 cm) [26]. Therefore, to achieve a high treatment temperature, researchers often increase the laser power or dosage of PAs. However, the American National Standards Institute (ANSI) has established standard tolerance threshold values for the clinically safe use of PTT on the skin [27]. The 808 nm laser power threshold ranges from 330 to 350 mW cm^−2^ with an exposure time of 10–1000 s. Moreover, PTT inevitably damages normal tissue around the tumor site and leads to in vivo toxicity and side effects [28]. Furthermore, several cell contents and some residual tumor cells caused by thermal ablation may cause a series of side effects, including inflammation, tumor metastasis, harm to normal tissues, and tumor recurrence [29].

To circumvent these limitations, mild-temperature photothermal therapy (MTPTT), with a temperature range from 42 °C to 45 °C [3,30], was introduced to reduce the temperature used, thus alleviating the side effects. In addition, MTPTT does not significantly affect the quality of life of the patient owing to the milder temperature used. However, MTPTT is associated with poor therapeutic effects. Therefore, studies have been exploring methods to achieve better therapeutic efficacy using nanocarriers under MTPTT. Though heat shock protein (HSPs) inhibitors or other compounds can be encapsulated into the nanoplatforms, the antitumor efficacy and safety still need more comprehensive and in-deep studies. Nevertheless, it is a significant integrative treatment and exhibits great potential in future clinical applications [31].

The current review comprehensively summarizes the recent advances and functions of novel nanosystems comprising MTPTT for the treatment of tumors (Figure 1). The review explores (1) the mechanism of action of MTPTT, (2) diverse approaches of MTPTT, (3) the combination of MTPTT with other therapeutic modalities, (4) challenges and future development of MTPTT to provide a basis for improving the efficacy of MTPTT. The studies explored the role of heat shock response (HSR) in the efficacy of MTPTT and how they can be used to improve the efficacy of MTPTT by designing drug delivery nanosystems. Finally, the current crucial challenges faced in the MTPTT field are explored, and some considerable future research directions are proposed to improve the existing strategies and to lay a basis for developing new strategies to improve the effectiveness of MTPTT.

## 2. The Mechanism of MTPTT

MTPTT effectiveness in cancer treatment does not depend on precise devices or special methods to control the mild temperature but on methods for maintaining treatment efficacy at mild temperature. MTPTT therapeutic effect is attributed to damage to the self-protective mechanism of tumor cells and preventing serious damage from heat stress. Studies report that MTPTT exerts its activity through two self-protective mechanisms, including heat shock reaction and autophagy [32,33]. In conventional PTT (>48 °C), thermal ablation induces severe and irreversible denaturation of proteins, DNA damage, and denaturation, and destroys the effective defense of the self-protective mechanism. Notably, the self-protective mechanism has a significant effect on the repair of unfolded proteins in MTPTT (<45 °C). Therefore, inhibiting the pathway of the self-protective mechanism is the most effective way to achieve the high efficacy of MTPTT. Studies report that HSR and autophagy are key targets for mediating self-protective mechanisms during MTPTT (Figure 2).

Hyperthermia above 41 °C causes protein denaturation and temporary cell inactivation, which may last for several hours [34,35]. As a result, upregulation of expression of HSPs is induced by HSR, thus effectively preventing aggregation of other proteins. HSR is a cellular defense mechanism present in all organisms and plays a role in preventing damage from hyperthermia or other adverse stress conditions. HSR limits the therapeutic efficacy of MTPTT through its cytoprotective and antiapoptotic effects [36]. Moreover, HSPs can interact with apoptosis signaling pathway proteins to inhibit the occurrence of apoptosis, thus reducing the therapeutic effect of hyperthermia [37,38]. In addition, tumor cells overexpress HSPs compared with normal cells, which makes them less sensitive to heat treatment and enables them to remain active at high temperatures [39,40].

Tumor cells mainly regulate the expression of HSPs by activating heat shock transcription factors (HSFs) [41]. Previous studies have explored four HSFs, including HSF1, HSF2, HSF3, and HSF4. Notably, HSF1 is the main transcription factor that mediates HSR. HSF1 is a highly expressed protein in various tumor cells and is related to tumor progression and poor prognosis. The main mechanism of action of HSF1 is by enhancing phosphorylation of its own 326 site serine, thus upregulating expression of HSP70 and HSP27 and ultimately promoting malignant proliferation and apoptosis resistance [42,43]. Expression levels of HSPs are low, and only 1–2% of the total protein exists under normal physiological conditions [44]. HSF1 is activated and bound to the promoter region of the downstream HSPs gene to promote the expression of HSPs after stimulation by high temperatures, excessive reactive oxygen species (ROS), or inflammation. HSP70 is mainly the first expressed protein as a result of HSR in many HSP families [45,46]. B-cell lymphoma-2 (Bcl-2) associated athanogene 3 (BAG3) is the chaperone protein of HSP70 and can bind to the ATPase domain of HSP70 through the bag domain to modulate HSP70 function [47,48]. In addition, the BAG3-HSP70 complex can bind to Bcl-2 and protect it from degradation, thus inhibiting the apoptosis pathway or inhibiting tumor cell apoptosis induced by hyperthermia therapy and chemotherapy [49,50,51].

Therefore, inhibition of HSR can reduce the thermoresistance of tumor cells to increase the effectiveness of sensitizing PTT. Several studies have explored the inhibition of HSR by gene-mediated silencing technology (small interfering RNA or short hairpin RNA, siRNA, or shRN and A), studies are developing heat-sensitive drugs. The efficacy of MTPTT is mainly achieved by blockingHSR, and is mainly through two aspects, including (1) reducing the synthesis of HSPs from HSR [52], and (2) inhibiting the activity of HSPs [53]. The current research mainly focuses on the mechanism of HSPs in improving the efficacy of PTT. The efficacy of PTT can be improved through the following three ways: use of HSPs inhibitors, silencing HSPs gene by siRNA and reducing ATP synthesis. Therefore, it is important to combine HSPs inhibitors (or siRNA, ATP inhibitors) with PAs in the nanosystem, thus improving the sensitivity of tumor cells to heat [54].

Besides, autophagy as a cellular self-protective mechanism rapidly activates cancer cells to maintain energy production and offer recycled materials in response to hyperthermia stress. Autophagy-related tolerance also acts a crucial role in thermal resistance [55]. There are three types of autophagy identified according to different routes in which substrates eventually enter into the lysosomal lumen: microautophagy, chaperone-mediated autophagy, and macroautophagy (Figure 3). Damaged and denatured proteins and organelles are engulfed by autophagosomes, then degraded in the lysosome to provide energy, and macromolecular precursors, and can be recycled to sustain cellular metabolism [56,57,58]. Therefore, intercepting the autophagy pathway can improve the efficacy of MTPTT. Autophagy can be blocked by inhibiting (1) formation of autophagosome (3-methyladenine, wortmannin) [56], (2) fusion of autophagosome and lysosome (hydroxychloroquine, chloroquine, vinblastine) [59], and (3) degradation of autolysosome (pepstatin A) [60]. On the contrary, excessive autophagy does not protect cells but destroys homeostatic functions and induces autophagy-mediated cell death (ACD), known as type II programmed cell death [61]. The excessive autophagy activity far exceeds the degradation capacity of the autolysosome, resulting in the formation of micron vacuoles and degradation blockage [62]. When autophagy fails to stop effectively or is overstimulated, the autophagic activities cannot recycle the cancer cellular components and accelerate ATP depletion, which ultimately leads to cell death and further enhance the therapeutic efficacy of MTPTT. Therein, excessive autophagy is induced via cutting off the inhibition pathway of autophagy or using autophagy inducers, including carbamazepine, C2-ceramide, rapamycin, and xestospongin B/C [63].

## 3. Various Approaches to Improve the Efficacy of MTPTT

Thermal ablation (above 48 °C) directly induces necrosis in tumor cells, whereas the surrounding normal tissues are damaged by heat diffusion [34]. This implies that PTT has a high therapeutic effect. However, it is characterized by adverse effects. Reduction in the temperature reduces the efficacy of PTT the owing to thermoresistance of tumor cells [3]. Therefore, developing strategies for overcoming thermoresistance is important to promote the efficacy of MTPTT. The next section explores the mechanism of thermal tolerance and summarizes approaches for constructing multifunctional nanosystems to improve the efficacy of MTPTT.

### 3.1. Heat Shock Proteins Inhibitors

HSPs are mainly classified as HSP27 (~27 kDa), HSP40 (~40 kDa), HSP60 (~60 kDa), HSP70 (~70 kDa), HSP90 (~90 kDa) and HSP110 (~110 kDa) based on their molecular weight [44]. HSP70 and HSP90 play important roles in HSR [39,64,65,66]. HSPs have some similarities in structure and function. All HSPs classes comprise three domains (Figure 4A), including the N-terminal domain, intermediate domain, and C-terminal domain [42,44]. The N-terminal domain is the binding site of ATP. Proline residues in the ATPase domain can induce conformational changes and cause hydrolysis, thus inducing the activity of HSPs [43]. The intermediate domain is the binding site of a guest protein and chaperone protein and is the active region of HSPs [45]. The C-terminal domain is a binding site for chaperone protein and is responsible for substrate binding and refolding (the dimerization of HSPs), resulting in “blocked” conformation to protect the guest protein [67].

HSPs inhibitors (Table 1) can specifically bind to the intermediate domain of HSPs, preventing binding of the guest protein and thus losing the ability to protect cells during HSR [71]. Gambogic acid (GA) is a natural prenylated xanthone moiety isolated from *Garcinia hanburyi*, and it presents various biological activities, such as anticancer, anti-inflammatory, antioxidant, and antibacterial activities. In addition, GA plays an important role by binding to the N-terminal ATP-binding domain of HSP90 without competing with, ATP thus inhibiting the catalysis of ATP hydrolysis [72,73]. GA has used an inhibitor of HSP90 due to this function and is combined with PAs to improve MTPTT efficacy in cancer treatment. Smart nanosystems are designed to achieve rapid release in tumor tissues or cells, thus improving the efficacy of drugs targeting HSPs. Yang et al. [68] designed poly (ethylene glycol) (PEG)-modified one-dimensional indocyanine Green (ICG)-Mn nanomaterials loaded with GA for MTPTT (Figure 4B). The one-dimensional ICG-Mn nanomaterial has the advantages of a high loading rate and pH stimulation response. In the acidic microenvironment of the tumor, the structure dissociates and rapidly releases GA. The cell survival rate of the GA-loaded nanoparticles group in vitro was significantly less compared with that of other groups at ~43 °C. In addition, the Western blot test showed that GA downregulated the expression of HSP90. GA-loaded nanocarriers can induce effective apoptosis of tumor cells under relatively mild temperatures by inhibiting HSP90 rather than thermal ablation (above 50 °C). It helps minimize the non-specific thermal effects on normal organs and improves the efficacy of PTT treatment of large or deep tumors.

Notably, insufficient tumor tissue accumulation and excessive liver retention availably limit the curative effect and biocompatibility of plenty of nanomedicines. Wu et al. reported smart theranostic nanocarriers consisting of GA as an HSP90 inhibitor, dc-IR825 as a fluorescence imaging probe and photothermal agents, and biocompatible human serum albumin [74]. The nanocarriers showed the synergy of chemotherapy and MTPTT, thus improving the efficacy of cancer treatment. In the nanocarriers, cytosolic translocation of GA can be promoted through ROS-mediated mitochondrial disruption under near-infrared (NIR) laser irradiation, further blocking the overexpression of HSP90. Hence, the nanocarriers can kill cancer cells under MTPTT, thus improving effectiveness in cancer treatment.

17-allylamino-17-demethoxy-geldanamycin (17-AAG) is an HSP90 inhibitor derived from the geldanamycin antibiotic and can cause the apoptosis of tumor cells [75,76]. Moreover, 17-AAG can effectively inhibit several cell signals transduction pathways, such as decreasing cellular levels of serine/threonine kinase 38 (STK38)/nuclear Dbf2-related 1 (NDR1) and the activity of STK38 kinase [77]. Wu et al. prepared hollow mesoporous organosilica nanocapsules (HMONs) to provide a versatile nanoplatform for imaging-guided MTPTT/chemotherapy, thus achieving high theragnostic efficacy (Figure 4C) [69]. 17AAG and ICG were loaded onto HMONs and then simultaneously released when the gemcitabine (Gem) gatekeeper was specifically open because of hydrolysis of the acetal bonds at acid TME. Then, 17AAG induces downregulation of HSP90 and thus reverses the thermoresistance of tumor cells to achieve the aim of MTPTT. This nanoplatform with, exhibiting the synergistic effect of MTPTT/chemotherapy, is the potential for precise cancer theranostics.

Quercetin (Qu) is a natural polyphenol-rich in hydroxyl groups widely distributed in vegetables, fruit peels, seeds, beverages, and Chinese herbs. It exhibits excellent antioxidant, anticancer, prevention, and treatment of cardiovascular and cerebrovascular diseases. Moreover, Qu inhibits HSP70 expression by regulating HSF transcriptional activity [78,79]. Yang et al. designed novel Qu-Fe^Ⅱ^P nanoparticles using quercetin, an HSP70 inhibitor (Figure 4D), as the framework [70]. High temperatures can induce inflammation and damage normal cells around tumor tissue. Therefore, quercetin plays an important role in clearing ROS and presents anti-inflammatory activity. The IC50 of Qu-Fe^Ⅱ^P on MCF-7 cells was 100 μg/mL in vitro. However, the IC50 of Qu-Fe^Ⅱ^P under laser irradiation was 3.13 μg/mL. After 20 days of treatment in vivo, the Qu-Fe^Ⅱ^P + laser irradiation group showed a good tumor inhibition effect under MTPTT, and three-quarters of the tumors disappeared completely. Western blot analysis of tumor tissue showed that expression level HSP70 was lower compared with other proteins, thus significantly reducing the thermal tolerance of tumor cells and improving MTPTT efficacy.

### 3.2. siRNA

RNA interference (RNAi) technology has great potential in the biomedical field. It provides a novel approach for the design and development of new drugs owing to its high efficiency, high specificity, and mild toxicity. Rational design, precise chemical modifications, and nanocarriers provide available opportunities to overcome the limitations of siRNA, such as rapid degradation, poor cellular uptake, and off-target effects. siRNA is an effective vector for RNA interference which inhibits the expression of HSPs or BAG3 by abrogating the expression of specific genes, making cancer cells more vulnerable to PTT effects, thus providing a strategy for inhibition of HSR [54]. However, the approach is characterized by limitations such as low serum stability and elimination in vivo during siRNA delivery to target cells. Therefore, its effectiveness can be improved by using nanosystems to deliver siRNA. Ding et al. [34] used siRNA as a crosslinking agent to construct DNA-grafted polycaprolactone (DNA-g-PCL) nanoparticles (Figure 5A) and then packaged them using polydopamine (PDA) and modified them using PEG (PP-NG-siHSP70). The gene silencing activity of PP-NG-siHSP70 with laser irradiation group presented the lowest expression of HSP70 mRNA and the highest expression of caspase-3 mRNA in agreement with the western blot results. The cellular apoptosis rate for the PP-NG-siHSP70 laser irradiation group in vitro was 72.2%, which indicated that the nanoparticle-induced effective target gene knockdown and apoptosis under mild conditions rather than cell necrosis. Notably, two-thirds of tumors in the PP-NG-siHSP70 laser irradiation group disappeared after 16 days of treatment. Wang et al. [82] prepared a gold nanorod (GNRs) platform loaded with BAG3 siRNA with gene silencing ability to improve the efficacy of MTPTT (Figure 5B). The findings using oral squamous cell carcinoma in vitro and in vivo showed that the nanorod improved the sensitivity of tumor cells to PTT and increased apoptosis after through downregulation expression of BAG3. The relative volume of GNRs-siRNA in mice treated with siRNA decreased by 18.4% after laser treatment.

### 3.3. Nucleus Damage

PAs target the nucleus, resulting in the destruction of the structure of genetic material in the nucleus, thus improving the efficacy of MTPTT. TAT is a cell-penetrating peptide that exhibits its function by targeting the nucleus. Thus, it can be used to target the nucleus by modification of ultra-small nanoparticles or quantum dots [85]. Therefore, the design of novel nuclear targeting PAs with efficient photothermal conversion properties and high intranuclear accumulation is significantly desired for MTPTT. Cao et al. [83] prepared small-size vanadium carbide (V_2_C)-TAT quantum dots that can accumulate nucleus and destroy genetic material, thus enhancing the effect of MTPTT (Figure 5C). The V_2_C-TAT quantum dots were coated by exosomes modified with RGD (V_2_C-TAT@Ex-RGD), which has a long-life cycle, high biocompatibility, and good tumor-targeting ability. V_2_C-TAT@Ex-RGD has a significant therapeutic effect in vivo owing to long blood circulation time, strong targeting ability of tumor cells, and less tumor accumulation under irradiation with a 1064 nm laser with a power density of 0.96 W cm^−2^. Therefore, the V_2_C-TAT@Ex-RGD can achieve nuclear targeting for guiding by multimodal imaging at mild temperature, which shows a good prospect for biomedical, and clinical application.

In addition to the cell-penetrating peptide-mediated nuclear target, ultrasmall nanoparticles are more likely to enter the nucleus through the nuclear pore (~40 nm size) and nuclear pore complex [86]. Liu et al. [87] prepared special ultrasmall chitosan-coated ruthenium oxide nanoparticles (CS-RuO_2_ NPs) with a nuclear target for MTPTT application in cancer in the near-infrared window and synthesized RuO_2_ NPs using a simple one-pot method (Figure 5D). Analysis of the nuclear power source with different sizes and surface charges showed that only the nuclear power source with ultrasmall size (2 nm) and positive charge could help effectively enter the nucleus destroying DNA and protein. The CS-RuO_2_ NPs revealed strong absorption in the NIR-II window with great photothermal conversion efficiency. In addition, they are ideal materials for PAs and photoacoustic imaging (PAI).

### 3.4. Energy Inhibition

HSPs are ATP-dependent proteins and are synthesized in large quantities under the presence of ATP [45,47,67]. Therefore, HSPs levels can be reduced by limiting the availability of energy. The main pathways for obtaining energy in tumor cells are glutamine metabolism, glycolysis, and autophagy and not oxidative phosphorylation of normal cells [88,89]. Therefore, these pathways can be regulated to limit the production of ATP in tumor cells. Starvation therapy aims to inhibit tumor cells’ access to or utilization of nutrients so that they can “starve to death” due to lack of energy [90]. Zhou et al. [29] used the catalysis of glucose oxidase (GOx) to oxidize glucose to gluconic acid and H_2_O_2_, thus limiting the utilization of glucose by tumor cells (Figure 6A). Mesoporous hollow PB NPs loaded glucose oxidase was designed based on this mechanism for a combination of starvation therapy and MTPTT for tumor treatment. In addition, PB NPs were used to catalyze H_2_O_2_ to improve the hypoxia level in tumor tissue. Starvation therapy limits the supply of ATP and inhibits the synthesis of HSPs, thus reducing the heat tolerance of cancer cells. The findings showed that the intracellular oxygen concentration decreased from 5.1 mg/mL to 0.04 mg/mL under the catalysis of GOx for 10 min, indicating that GOx catalyzed glucose oxidation. The tumor volume decreased by 32.5% after 21 days of treatment using synergistic therapy.

In another attempt to address the toxic side effects of the high dose of HSP inhibitor, Gao et al. demonstrated a thermosensitive GOx/indocyanine green/gambogic acid (GA) liposomes (GOIGLs) for enhancing the efficiency of MTPTT via synergistic inhibition of HSPs from GA and GOx which induced glucose consumption via catalyzing glucose into gluconic acid (Figure 6B), together with enzyme-improved phototherapy effect [91]. In addition, H_2_O_2_, as the product of the oxidation of glucose, can be converted into hydroxyl radical (·OH) under light irradiation, which effectively eliminates cancer cells to realize enzyme-enhanced phototherapy (EEPT). From the results of cancer cells and tumor-bearing mice experiments, the significant antitumor efficacy of “GOIGLs + Laser + Light” demonstrated that GOx-mediated tumor starvation and phototherapy improved the therapeutic efficiency of MTPTT. Compared with conventional PTT, MTPTT not only can achieve an effective antitumor therapy at a relatively low temperature (below 45 °C) but also reduce thermal damage to the normal tissues from the safety evaluation results.

Consuming a large amount of glucose is one effective way to burn energy, hindering the efficiency of HSPs. In addition, glucose uptake is also a vital target for inhibiting the metabolic pathway of cancer cells. Chen et al. [92] used diclofenac to inhibit the activity of glucose transporters (Gluts) (Figure 6C), thus limiting the uptake of glucose by tumor cells. It achieved the purpose of MTPTT by downregulating the synthesis of HSP70 and HSP90 after reducing the anaerobic glycolysis of tumor cells, thus reducing the level of ATP. Western blot analysis presented that the amount of Glut1 protein decreased significantly after HeLa cells and MCF-7 cells were cultured with GNR/HA-DC for 12, 24, and 48 h. GNR/HA-DC caused a significant reduction in glucose uptake by cancer cells, which inhibited cell functions, including anaerobic glycolysis. ATP decreased by 52.7% and 35%, respectively, after 48 h of culture.

### 3.5. Autophagy Mediation

Autophagy, as a dynamic cellular pathway, degrades and recovers damaged or aged proteins and organelles. Dysfunctional autophagy is related to cancer, microbial infection, neurodegeneration, and aging, indicating that autophagy plays a key role in these diseases [61]. Several studies report that drug inhibition of autophagy or gene knockout of autophagy-related genes (ATG) can increase the sensitivity of cancer cells to a variety of drugs [93,94]. Chloroquine (CQ) inhibits autophagy and enhances the anticancer activity of histone deacetylase inhibitors against chronic myeloid leukemia [95]. Moreover, inhibition of autophagy can enhance the anticancer effect of bevacizumab against hepatocellular carcinoma. Besides, CQ is also applied for the treatment of malaria and autoimmune diseases [96].

Several studies have explored the modulation of autophagy for the development of drugs for cancer treatment. The heat transformed by the photothermal effect in PTT activates autophagy by damaging cytoplasmic components. Therefore, inhibition of autophagy can improve the therapeutic effect of MTPTT. CQ inhibits the degradation of autophagy by inhibiting lysosomal activity. Zhou et al. used CQ to inhibit the autophagy of tumor cells, thus improving the efficacy of MTPTT (Figure 7A) [97]. The findings showed that the tumor volume of the PDA-PEG/CQ group was about 30 mm^3^, and the temperature of laser irradiation was controlled at ~42 °C.

Moderate autophagy helps cells survive in an adverse environment; however, excessive autophagy leads to cell death. Unlike apoptosis, autophagy is characterized by the forming of a large number of autophagosomes wrapped in cytoplasm and organelles. Beclin1 coupled polymer nanoparticles promote autophagy activity in tumor cells and further inhibit the growth of the tumor. Beclin1 induced autophagy abrogates the homeostasis function of autophagy, activates the autophagy cell death pathway, and improves the therapeutic effect of PTT. Zhou et al. [98] prepared multifunctional nanoparticles for tumor targeting and for improving PTT efficacy through autophagy induction (Figure 7B). The nanoparticles comprised PDA nanoparticles and Beclin 1-derived peptide, PEG, and cyclic Arg-Gly-ASP peptide (PPBR). PPBR improved the autophagy activity of cancer cells and significantly promoted the killing efficiency of PTT. Findings from animal experiments showed that PPBR could upregulate autophagy of tumor cells, and the combination therapy inhibited tumor growth more effectively compared with single therapy in a breast tumor model.

## 4. The Synergistic-Therapy Strategies

Nowadays, many preclinic and clinic researchers have demonstrated that many monotherapies show low efficiency in curing tumors, high recurrence rate, severe toxic side effects like hematologic toxicity, abnormal liver function, and high toxicity to normal cells, and immune system disorder. The limited penetration depth of light lowers the efficacy of PTT in inhibiting tumor growth outside the radiation range [99]. Monotherapies are inefficient in abrogating tumor growth, including PTT monotherapy. Although PTT has a high therapeutic effect, its limitations may contribute to the incomplete elimination of tumor cells, ultimately leading to tumor recurrence and metastasis. The combination of MTPTT with other treatments can improve therapeutic efficacy. In addition to providing a supplement, the combination of different therapeutic approaches results in a synergistic treatment effect.

### 4.1. Chemotherapy

MTPTT improves the therapeutic effect of chemotherapy through several mechanisms, including (1) increasing toxicity of some drugs, (2) increasing uptake of nanoparticles by tumor cells, (3) stimulating the rapid release of drugs from nanoparticles, and (4) increasing sensitivity of multidrug-resistant cells to chemotherapy. In addition, chemotherapy can kill the metastatic tumor cells, whereas MTPTT does not eliminate metastatic cells. Therefore the combination of MTPTT and chemotherapy exhibits a good synergistic effect on tumor treatment [13,100]. Fu et al. [81] designed novel multifunctional boron-based nanoplatforms combining chemotherapy and MTPTT. The boron-based nanoplatforms targeted αvβ3 integrin overexpressed in tumor cells through functionalization with cyclo (Arg-Gly-Asp) (cRGD) peptide (Figure 8A). DOX (603 mg g^−1^) and 17AAG (417 mg g^−1^) were loaded with boron nanosheets. The DOX-17AAG@B-PEG-cRGD systems exhibited controlled, and NIR induced DOX and 17AAG release. DOX-17AAG@B-PEG-cRGD systems significantly enhanced the cellular uptake of cancer cells compared with healthy cells. The presence of 17AAG in a combination of MTPTT and DOX chemotherapy improves anticancer activity. These multifunctional nanoplatforms are promising candidate platforms for tumor therapy. Wu et al. [69] designed HMONs loaded with ICG and 17AAG, and the antitumor drug gemcitabine was modified through a pH-sensitive acetal covalent bond to block the pore. Furthermore, NH_2_-PEG can be introduced through the benzamide bond, which improves the cycling performance of the nanoparticles. ICG-17AAG@HMONs-Gem-PEG nanoparticles exhibit pH-responsive molecular release and glutathione (GSH) dependent degradation in TME. ICG and 17AAG can be released on demand owing to hydrolysis of the acetal bond under weak acidity (<6.0). Subsequently, 17AAG regulates HSP90, thus abrogating the heat tolerance of tumor cells. In addition, it can effectively induce cancer cell apoptosis under relatively mild-temperature PTT. Gemcitabine acts as a gatekeeper and can be released from nanocapsules as a chemical for cancer chemotherapy. The near-infrared fluorescence and PAI of nanocapsules present high contrast owing to the strong near-infrared absorption of ICG, which helps in targeted treatment.

### 4.2. Radiotherapy

MTPTT alone does not completely eliminate deep tumors owing to its limitations. RT can damage DNA and cause cancer cell death without depth limitation [101]. However, the degree of cell damage is significantly affected by the level of intracellular oxygen ions induced by ionizing radiation. Therefore, the hypoxia TME significantly limits the effectiveness of RT [2]. Notably, MTPTT induced hyperthermia can accelerate tumor blood flow and improves TME oxygen status, thus increasing the sensitivity of tumor cells to RT [102,103]. Therefore, the combination of RT and PTT is a promising strategy for tumor eradication, which shows advantages such as improving treatment efficacy and reducing side effects. In addition, hyperthermia can effectively inhibit the repair of nonlethal X-ray injury, resulting in a significant synergistic effect of MTPTT/RT. Song et al. [104] developed Bi_2_Se_3_ hollow nanocubes (HNCs) modified with hyaluronic acid loaded with GA (HNC-S-S-HA/GA) (Figure 8B). Downregulation of HSPs mediated by GA reduces the resistance of cancer cells to heat stress. HNC-S-S-HA/GA effectively induces apoptosis of cancer cells and has a significant ablation effect on cancer cells. The heat generated by light and heat increases blood flow resulting in the delivery of more oxygen into cancer cells, thus alleviating the hypoxic TME. HNC-mediated enhanced RT showed an improved cancer cell-killing effect under X-ray irradiation. Novel HNC-S-S-HA/GA nanocubes have several unique advantages such as (1) high stability, drug loading capacity, and absorption coefficient, (2) HNC-S-S-HA/GAS has enhanced permeability and retention effect/CD44 mediated bimodal tumor targeting and GSH sensitive drug release, further reducing toxicity to normal cells, (3) MTPTT can inhibit proliferation of surrounding tissues, (4) inhibition rate of MTPTT combined with RT is significantly higher compared with that of MTPTT or RT alone. Novel HNC-S-S-HA/GA-based TME responsive nanodrugs show potential for use in mild-temperature MTPTT/RT guided by multispectral optoacoustic tomography (MSOT)/computed tomography (CT) imaging, with high anti-tumor efficiency and minimal invasion in normal tissues.

### 4.3. Photodynamic Therapy

PDT is a therapeutic approach mainly using photosensitizers (PSs) to transfer energy to oxygen to form highly active singlet oxygen (^1^O_2_) under laser irradiation, selectively destroying cancer cells by oxidating the proteins, nucleic acids, and lipids (Type II mechanism) [31,105,106]. In addition, PSs produce ROS, such as a hydroxyl radical or a superoxide anion, which is the Type I mechanism [107]. Due to its advantages of being non-invasive, high safety, spatiotemporal control, and broad spectrum, PDT has attracted much attention and is considered a potential tumor treatment. However, one of the biggest bottlenecks in Type II PDT is hypoxic TME, which severely decreases the ^1^O_2_ yield. MTPTT significantly improves tumor oxygen supply by increasing the blood flow, which is a benefit to enhancing the singlet oxygen generation efficiency. Moreover, PDT interferes with tumor physiology and microenvironment and enhances the thermal sensitivity of tumor cells. The combination of MTPTT and PDT exhibits several advantages in improving the effectiveness of tumor therapy [108].

Hence, a variety of nanosystems have been developed for MTPTT/PDT synergistic therapy [109,110]. Notably, GQDs-based nanocomposites for PDT can act not only as PSs but also nanoplatforms for delivering PSs [111]. In addition, some inorganic nanomaterials also become PSs and delivery systems. However, most of these systems require complex integration or assembly components to achieve an “integrated” function [112,113]. Studies are currently exploring the use of organic metal frameworks (MOFs) in PDT/PTT and multimode imaging to reduce additional integration of other components. PSs can be directly used as a component (linkers) of MOFs to achieve non-invasive PDT of tumors [114]. However, the production of ROS limits therapeutic effect, mainly because the single PDT treatment does not present an effective anti-cancer effect [115,116]. Therefore, there is a need to develop multifunctional MOFs to improve diagnostic accuracy and treatment efficacy. Zhang et al. [117] developed new multifunctional zirconium-ferriporphyrin MOF (Zr-FeP-MOF) nanoshuttles (Figure 8C). HSP70 inhibitor siRNA was modified using PEG and loaded with siRNA to prepare the siRNA/Zr-FeP-MOF therapeutic platform. Notably, the siRNA/Zr-FeP-MOF can catalyze endogenous hydrogen peroxide (H_2_O_2_) and O_2_ to generate abundant hydroxyl radical (·OH) and ^1^O_2_ under NIR laser. siRNA/Zr-FeP-MOF nanoswitches have a photothermal effect on MTPTT owing to the introduction of siRNA. It indicated that PDT combined with MTPTT could significantly inhibit cancer growth in vitro and in vivo.

### 4.4. Gas Therapy

Organisms have gaseous signaling molecules that act as messengers in the process of specific binding with multivalent transition metals like hydrogen sulfide (H_2_S), nitric oxide (NO), and carbon monoxide (CO) [118], and thus have a variety of physiological functions in almost all human systems, such as the nervous system, cardiovascular system, and immune system. The gas signaling molecules play an important role in normal human physiological processes and are implicated in the regulation of pathological processes. A high concentration of NO, CO and H_2_S in the blood can cause poisoning; however, in a relatively mild concentration range, they have significant anti-cancer activity [119]. For instance, NO mainly reacts with the superoxide anion to form reactive nitrogen species (RNS), such as peroxynitrite, that react with DNA to induce a variety of DNA damage [120]. The efficacy of gas therapy is highly correlated with gas concentration. Therefore, the controlled release of therapeutic gas in the lesion is very critical in gas therapy [121,122].

Laser is the most convenient and effective exogenous stimulus, therefore, light-controlled drug release has been explored in several studies [123,124]. The penetration depth of ultraviolet and visible light significantly limits the application of light-responsive gas release in vivo and can easily cause phototoxicity. On the contrary, NIR light has better tissue penetration depth and milder phototoxicity. Therefore, NIR laser-responsive gas release has a broader application prospect. Laser irradiation ensures remote control of gas therapy, and the thermal effect of PTT can promote the release of the gas [125,126,127]. For instance, Gao et al. utilized PAs to combine with a photo-triggered NO generator (thiolated transferrin), thus promoting the release of NO under the irradiation of 808 nm near-infrared light [128].

However, the application of NO in biomedicine is limited by several aspects, such as high activity, poor selectivity, and short half-life (less than 3s). Therefore, selective delivery of NO based on nanoplatforms to the lesion area should be explored to ensure maximum utilization of NO in anti-cancer therapies. The thermal effect of MTPTT promotes a spatiotemporally controlled release of NO. Yao et al. [80] designed gold nanorods coated with mesoporous silica, linked with S-nitrosothiols, and loaded them with 2-phenylethynesulfonamide (PES) as an inhibitor of HSP70, and surface modification using PEG (Figure 8D). The cumulative concentration of NO reached 14.6 μM after 10 min of irradiation with 1 W/cm^2^ 808 nm laser, and the solution temperature was 40.8 °C. The apoptosis rate of the combination therapy comprising gas therapy and MTPTT was approximately 70%, which was two-fold that of MTPTT alone. The tumor inhibition rate of the combination of gas therapy and MTPTT was approximately 85% in vivo.

**Figure 8 pharmaceutics-14-02279-f008:**
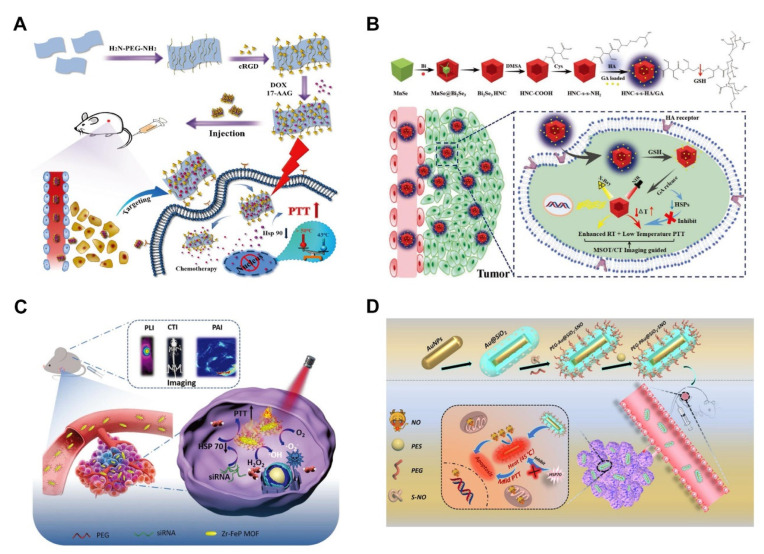
A variety of examples reveal the superiority of the synergistic therapy MTPTT with other therapies. (**A**) A schematic illustration of the preparation of DOX-17AAG@B-PEG-cRGD nanosheets and the synthetic approach of MTPTT with chemotherapy [81]. Copyright © 2022, Royal Society of Chemistry. (**B**) Illustration of synthesis procedure of HNC-s-s-HA/GA and the combination of MTPTT with radiotherapy [104]. Copyright © 2022, John Wiley and Sons. (**C**) Schematic illustration of siRNA/Zr-FeP MOF nanoshuttles for multimode imaging diagnosis and combination of MTPTT and PDT for cancer treatment [116]. Copyright © 2022, John Wiley and Sons. (**D**) Schematic illustration of the preparation of PEG-PAu@SiO_2_-SNO nanocomposites and the process of mild heat-enhanced gas therapy under NIR irradiation in MCF-7 cells [80]. Copyright © 2022, American Chemistry Science.

### 4.5. Immunotherapy

Cancer immunotherapy is the most promising and epochal treatment that boosts anti-cancer immunity or eliminates immune suppression to achieve immune cell-mediated tumor clearance and improve the survival of cancer patients [129,130,131]. Normally, the immune system recognizes and kills abnormal tumor cells, but tumor cells can trigger a large of strategies to avoid recognition and elimination by the immune system through a process known as “immune escape” [132]. Cancer immunotherapy is a treatment to eradicate tumors by restarting and maintaining the cancer-immunity cycle [133]. The cancer-immunity cycle is divided into the following seven links: (1) the release of antigens from the dead cancer cells, (2) antigen presentation, (3) priming or activating T cells, (4) T-cell migration to tumors, (5) tumor tissues infiltrating T-cells, (6) T cells recognize tumor cells, (7) eliminating cancer cells. One of the barriers in any of these links can result in failure of the anti-tumor—immune cycle and immune escape. However, immunotherapy is not effective for all tumor types, owing to the toxicity of high immunohorizons and low objective rate [134].

Tumor recurrence is a severe defect caused by PTT due to residual tumor cells caused by uneven heating or failure to completely remove tumor cells due to the limitation of the laser penetrating depth [28]. In addition, PTT is an effective local tumor therapy. However, it is not effective for disseminated tumors. Recent studies report that high temperatures can promote the release of anticancer substrates from dead tumor cells, leading to immune activation [135,136,137]. Previous studies report that hyperthermia can trigger an immune response by activating immune cells such as CD8^+^ T cells, natural killer (NK) cells, and dendritic cells (DCs), promoting the release of tumor cells exons, and upregulating expression of inflammatory cytokines and HSPs [138]. PTT-induced hyperthermia can result in apoptosis or necrosis of tumor cells and release tumor-associated antigens. The tumor-associated antigens can be received and presented by antigen-presenting cells (APCs), thus activating immune cells, and inducing an anti-cancer immune response. Moreover, a combination of PTT and immune adjuvant can be used for the development of in situ autologous cancer vaccine [139,140].

Cancer vaccine is efficient immunotherapy that can effectively activate the immune system to kill tumor cells by injecting tumor-associated antigens into tumor patients so as to achieve the purpose of tumor control and treatment [141,142,143]. Tumor-infiltrating dendritic cells usually present an immature and immunosuppressive phenotype and are unable to mediate the immunosuppressive response of tumors fully. Immunotherapy adjuvants with CpG oligonucleotides as Toll-like receptor (TLR) agonists can activate tumor-infiltrating dendritic cells to enhance vaccine-specific immunity [144,145]. Nevertheless, the shortcoming of CpG oligonucleotide-based immunotherapy is usually counteracted by immunosuppressive TME [146,147,148,149]. For modulating the microenvironment toward immune activation, Li et al. prepared a photothermal CpG nanotherapeutics (PCN) (Figure 9A) to induce an immunofavorable TME by casting heat (43 °C) after laser irradiation in the tumor site [150]. The apoptosis results showed that an MTPTT from laser illumination could mediate tumor cell apoptosis and necrosis. The antigen released from tumor cells and CpG-activated macrophages and DCs to promote activation/maturation demonstrated from the results of the increased serum IL-6 level, the upregulated expression of BMDC maturation markers, CCL8 and Clec4e. As a results, MTPTT was successfully proven to improve the innate and adaptive immune response.

HSP90 inhibition improves tumor immunotherapy by upregulating the expression of the interferon response gene. HSP90 is highly correlated with autophagy and protein kinase B (AKT). However, severe side effects and tumor recurrence limits traditional treatment. Thus, it is challenging to obtain a satisfactory survival rate. Treatment can be more effective and milder by targeting the specific region or functions of cancer. “Automatic treatment” widely represents the “double-edged sword” phenomenon. On the one hand, autophagy plays a significant role in cancer eradication due to providing nutrition and limiting T cell-mediated cytotoxicity. On the other hand, the accumulation of autophagosomes can exert anti-tumor activity by inducing is associated with ACD. Therefore, drug-induced or inhibition of autophagy can kill tumor cells. Tumor cells produce a stress protein (HSPs) when exposed to laser irradiation to protect themselves through the normal function of HSPs from the invasion of MTPTT. On the contrary, autophagy can lead to cancer cell apoptosis when HSP90 is downregulated, which plays a role in several key ways to maintain the dynamic stability of the cell environment. This implies that the function of autophagy may be reversed in the absence of HSP90 when facing a high-energy environment. The application of MTPTT to regulate autophagy is a novel approach to tumor therapy. Overactivation of autophagy induced by MTPTT and regulation of HSP90 inhibitors play key roles in the efficacy of MTPTT [33]. Deng et al. designed graphene oxide (GO) loaded with SNX-2112 and folic acid (FA) for MTPTT (Figure 9B) [32]. Changes in HSPs levels are related to the activity of AKT because HSP90 is an early protein in the AKT signaling pathway. During autophagy, HSP90 inhibits AKT and inactivates AKT under stress conditions. Therefore, the pathways associated with vital signs were evaluated by Western blot analyses. The results showed that expression of level p-AKT and autophagy-related genes were significantly different in the GO-folic acid- SNX-2112 group and expression of HSP90 was inhibited by SNX-2112. These findings indicated that autophagy is activated and the AKT pathway is inhibited by MTPTT. In addition, the expression of programmed death-ligand 1 (PDL1), which is implicated in tumor immune function, was significantly downregulated compared with the level of the control group and GO-folic acid group. These findings indicate a relationship and crosstalk among autophagy, p-AKT, and PDLL.

Immune checkpoint therapy that aims to regulate the activity of T cells through inhibition or stimulation of signals from the immunosuppressive TME is the main treatment of immunotherapy [151,152]. However, recent research presented that many patients presented with “non-immunogenic” tumors, also known as “cold” tumors, characterized by a lack of tumor-infiltrating lymphocytes [153,154]. Therefore, it is a grand challenge for immunotherapy how to convert “cold” tumors to “hot” tumors. There are five strategies to switch cold tumors to hot tumors [155]: (1) improve the tumor inflammation, (2) neutralize the immunosuppressive factors in the TME, (3) target the tumor blood vessels and stroma, (4) target tumor cell signaling pathways, (5) improve the longevity and function of anti-tumor immune T cells. The combined therapy with various therapeutic treatments has enhanced the immune checkpoint therapy efficacy [156,157]. Moreover, the mild temperature is favorable to immunological responses in TME [115,158]. Huang et al. loaded a photothermal agent (IR820) and an anti-programmed death-ligand 1 antibody (aPD-L1) into a lipid mixture for combining the immune checkpoint blockade antibodies with MTPTT (Figure 9C) [159]. After measuring the immune response of immune cells in lymph nodes, spleens, and tumors of 4T1 and B16F10 mice, the results showed that the treatment-induced cell differentiation of naive T cells to CD8^+^ T cells. Therefore, the precise control of MTPTT temperature is critical in sensitizing the immunosuppressive tumors for converting cold tumors to hot tumor as well as potentiating immune checkpoint therapy.

## 5. Summary and Perspectives

Over the past decades, PTT of tumors has drawn extensive attention, and studies demonstrate broad prospects for small, unresectable tumors or for patients who are poor surgical candidates. High-temperature thermal ablation by agent-free PTT or contrast-enhanced PTT significantly affects healthy tissues, induces undesirable inflammation, and largely impairs immune antigens and immune cells related to antitumor immunity. Therefore, MTPTT was introduced to circumvent the side effects associated with PTT at a temperature below 45 °C. Studies report that a combination of MTPTT with HSPs inhibitors or other various agents significantly improves the efficacy of mild hyperthermia as these agents alleviate thermal resistance. In addition, well-designed and multifunctional nanoplatforms can improve tumor specificity of photosensitizers improving tumor targeting, selectivity, activation, or image guidance, and/or through combination with other therapies.

Nanosystems can achieve the high therapeutic efficiency of MTPTT by restricting the function or reducing the expression of HSPs. In the current review, the mechanism and recent strategies in multifunctional nanoformulations for enhanced MTPTT were summarized. The use of HSPs inhibitors, integrating siRNA, targeting the nucleus, blocking energy inhibition, and impacting autophagy can alleviate the thermal resistance of tumor cells and protect normal cells from MTPTT effects. In addition, synergistic therapy can combine the advantages and offset the disadvantages of individual therapies, thus improving therapeutic outcomes. The current review explored the recent developments in combining MTPTT with other therapies, including chemotherapy, radiotherapy, PDT, gas therapy, and immunotherapy. Notably, MTPTT can modulate and rebuild the immunosuppressive tumor environment from “cold” to “hot,” thus preventing tumor recurrence and metastasis. In summary, MTPTT has a wide clinical prospects and further advances in biomedicine compared with PTT.

Although previous studies have reported several multifunctional nanoagents and their positive outcomes, MTPTT is still in a nascent stage. The clinical application of MTPTT is limited by the poor penetration depth of laser light, accumulation at the target locations, and poor biocompatibility with nanomaterials in vivo. Further studies should explore NIR-Ⅱ PAs should be explored for orthotopic tumors. In addition, the use of nanoparticles is limited by undesirable immune reactions and rapid clearance from the body by the reticuloendothelial systems. Biomimetic and bioinspired coating can be used to overcome physical barriers, such as the use of cell membrane cloak nanomaterials. Broad particle size distribution of nanoparticles can result in high risks when used in humans. Biodegradable and naturally derived PAs can be used in MTPTT. Moreover, new technology or therapy modalities such as chiral nanoparticles, nanoenzymes, nanorobots, chemodynamic therapy, ultrasound therapy, and microwave therapy can be combined with MPPTT to improve therapeutic efficacies for MTPTT-based cancer treatment.

## Figures and Tables

**Figure 1 pharmaceutics-14-02279-f001:**
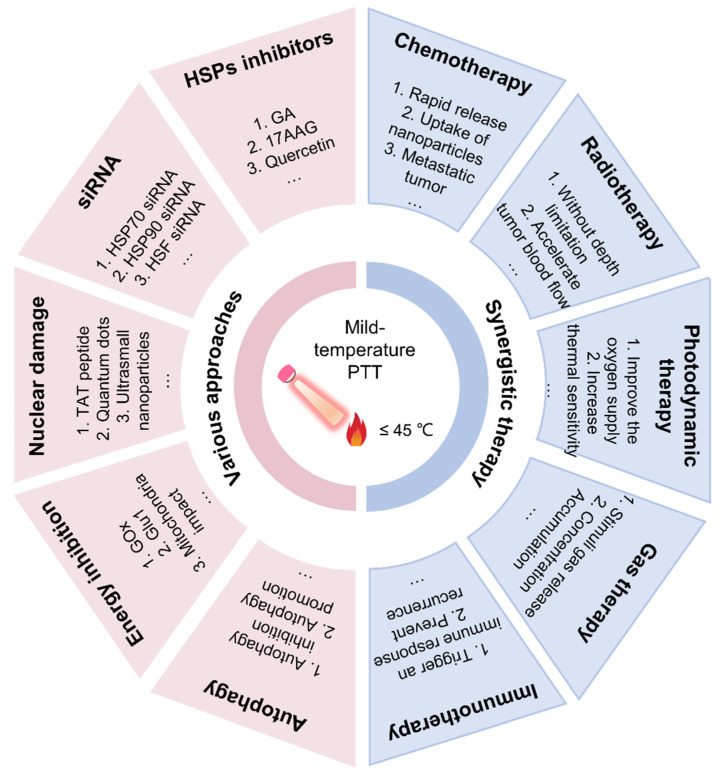
Scheme illustrating the use of MTPTT for cancer treatment via various strategies.

**Figure 2 pharmaceutics-14-02279-f002:**
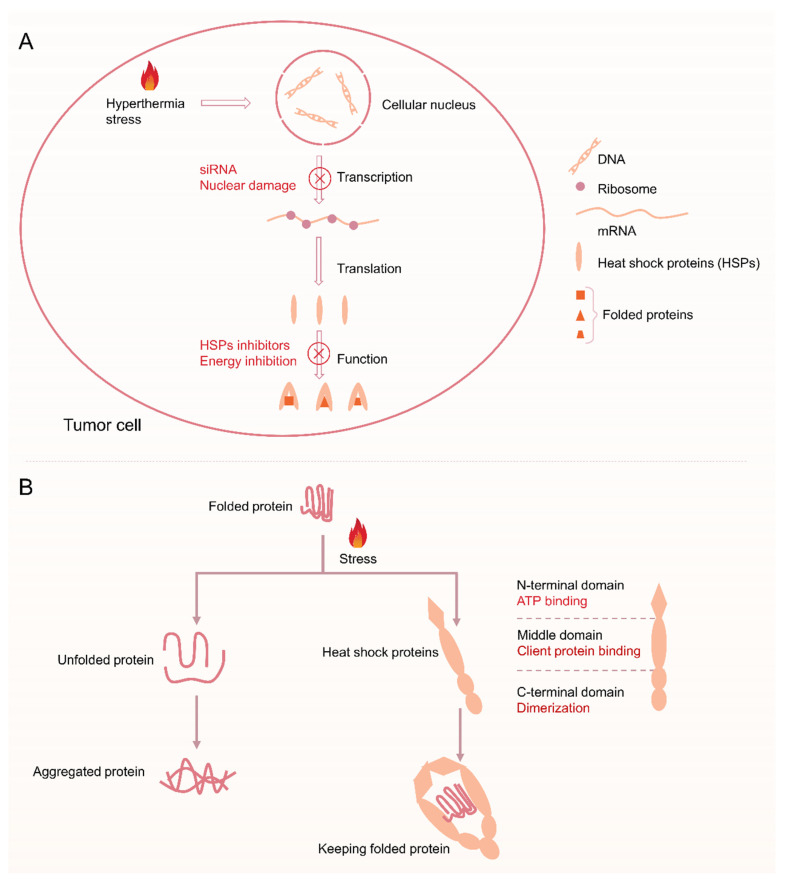
(**A**) Schematic of the process of heat shock reaction after hyperthermia and the blocking function of heat shock reaction via siRNA, nuclear damage, HSPs inhibitors, and energy inhibition. (**B**) Illustrating the physiological functions of HSPs: assists protein folding into its native form in MTPTT.

**Figure 3 pharmaceutics-14-02279-f003:**
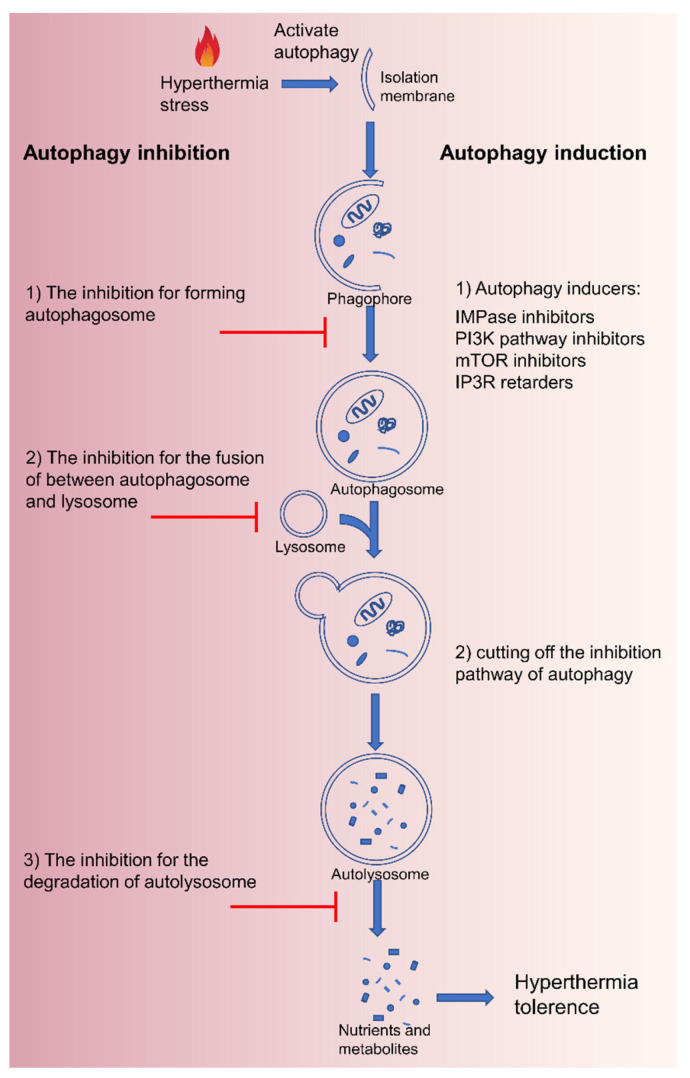
Schematic of the process of macroautophagy after hyperthermia and the various strategies to inhibit or induce autophagy.

**Figure 4 pharmaceutics-14-02279-f004:**
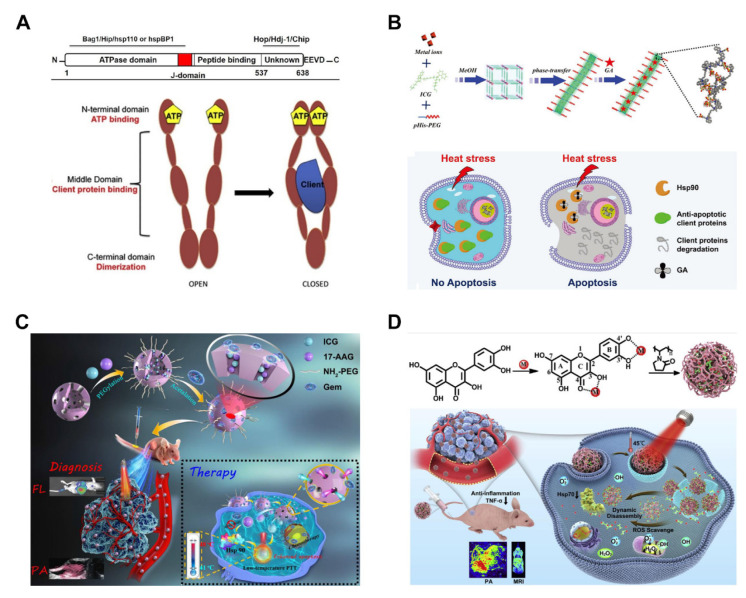
Several examples showing HSP70 and HSP90 inhibitors incorporated nanoplatforms that efficiently achieve MTPTT of the tumor. (**A**) Schematic representation of the structure and function of HSP70 and HSP90 [43,67]. Copyright © 2022 and 2016, Elsevier. (**B**) A scheme to illustrate a one-step synthesis of one-dimensional nanoscale coordination polymers and to overcome thermal resistance by inhibiting HSP90 [68]. Copyright © 2022, John Wiley and Sons. (**C**) Schematic illustration of the construction of the ICG-17AAG@HMONs-Gem-PEG nanoplatforms for fluorescence/photoacoustic imaging-guided MTPTT/chemotherapy [69]. Copyright © 2022, American Chemical Society. (**D**) Schematic representation of the one-pot synthesis of a family of poly(vinylpyrrolidone) protected metal ion-quercetin (Qu) coordination nanodrugs, intrinsically integrating precise diagnosis, excellent MTPTT efficacy, ROS elimination, and anti-inflammatory action, dynamic disassembly, and renal clearance ability into a single nanoparticle [70]. Copyright © 2022, Elsevier.

**Figure 5 pharmaceutics-14-02279-f005:**
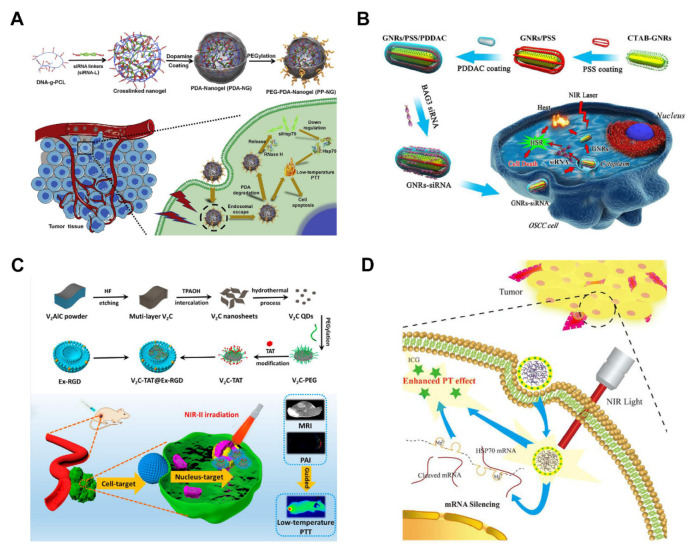
Specific means of siRNA and nucleus damage in favor of MTPTT. (**A**) Schematic illustration of the synthesis of PDA-coated nucleic acid nanogel and the mechanism of siRNA-mediated MTPTT induced by PEG-PDA-Nanogel [34]. Copyright © 2022, Elsevier. (**B**) Schematic illustration of the design of GNRs-siRNA in the improved MTPTT platform [82]. Copyright © 2022, Elsevier. (**C**) Schematic diagram of the cancer cell membrane and nucleus organelle dual-target V_2_C-TAT@Ex-RGD nanoagents for multimodal imaging-guided MTPTT in the NIR-II biowindow [83]. Copyright © 2022, American Chemical Society. (**D**) Illustration of DNAzyme-based nanosponges for highly efficient PTT [84]. Copyright © 2022, The Author(s).

**Figure 6 pharmaceutics-14-02279-f006:**
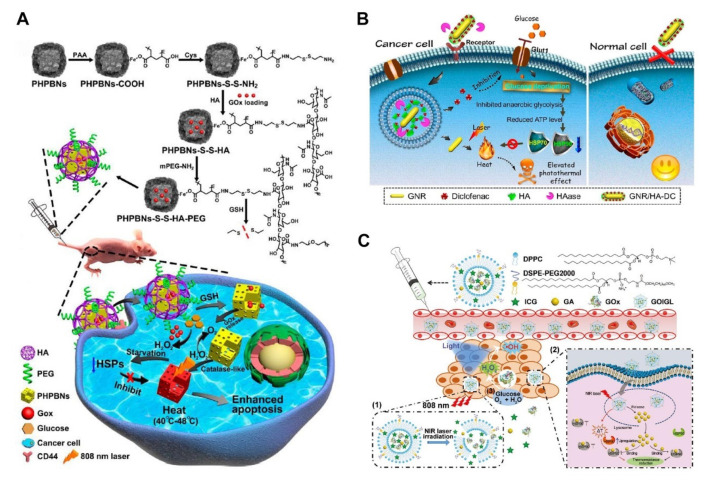
Strategies that enhance the therapeutic effect of MTPTT through inhibiting energy or metabolism. (**A**) Illustration of GOx-induced starvation for enhanced MTPTT in a hypoxic TME via the PHPBNs-mediated tumor reoxygenation [29]. Copyright © 2022, American Chemical Society. (**B**) Schematic illustration of thermosensitive liposomes encapsulating GOx, ICG, and GA for synergistic starvation therapy, EEPT, and enhanced MTPTT of tumors [91]. Copyright © 2022 John Wiley and Sons. (**C**) Schematic illustration of GNR/HA-DC for selectively sensitizing tumor cells to MTPTT by interfering with the anaerobic glycolysis metabolism [92]. Copyright © 2022, American Chemical Society.

**Figure 7 pharmaceutics-14-02279-f007:**
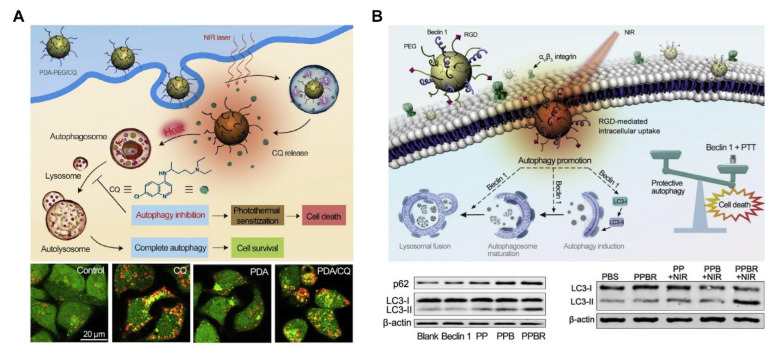
Two examples illustrate that autophagy sensitizes the photothermal killing of cancer cells. (**A**) Schematic of autophagy inhibition sensitizes photothermal killing of cancer cells. Western blot of LC3-I and LC3-II in HeLa cells treated with CQ and 3-MA as autophagy inhibitors, respectively [97]. Copyright © 2022, Elsevier. (**B**) The illustration depicts beclin 1-induced autophagy sensitizing photothermal killing of cancer cells. Western blot of LC3-I/LC3-II conversion and P62 in MDA-MB-231 cells treated with beclin 1, PP, PPB, and PPBR [98]. Copyright © 2022, Elsevier.

**Figure 9 pharmaceutics-14-02279-f009:**
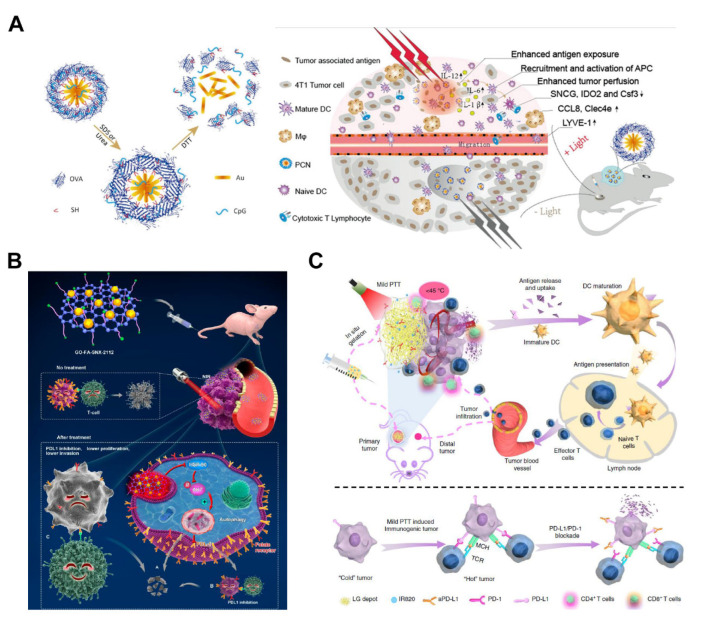
MTPTT assists in distinct types of immunotherapies. (**A**) The schematic of the disassembly process of PCN and an immunofavorable TME was established via a fever-like immune response induced by the photothermal effect of PCN [150]. Copyright © 2022, The Author(s). (**B**) Schematic structure of GO-FA-SNX-2112 and its application for MTPTT of the tumor to induce overactivation of autophagy; stimulated autophagy not only causes tumor cells to die directly but also makes them be captured by immunity because of the decrease in PDL1 receptor expression; residual surviving tumor cells are also gradually killed by restored immune cells, to achieve efficient inhibition of tumor growth [32]. Copyright © 2022, American Chemical Society. (**C**) Schematic illustration of the symbiotic mild photothermal-assisted immunotherapy via a combined all-in-one and all-in-control strategy [159]. Copyright © 2022, The Author(s).

**Table 1 pharmaceutics-14-02279-t001:** HSP inhibitors for MTPTT.

The Species of HSPs	Agents	PAs	Reference
HSP70	2-phenylethynesulfonamide (PES)	PEG-PAu@SiO2-SNO	[80]
	quercetin	Qu-Fe^II^P nanoparticles	[70]
		B780/Qu NPs	[78]
HSP90	17-AAG	ICG-17AAG@HMONs-Gem-PEG	[69]
		DOX-17AAG@B-PEG-cRGD	[81]
	Gambogic acid	NCPs/GA	[68]
		HAS/dc-IR825/GA	[74]

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
