# Peer review of "Enhancing the Efficiency of Mild-Temperature Photothermal Therapy for Cancer Assisting with Various Strategies"

_pharmaceutics, 2022, doi:10.3390/pharmaceutics14112279_

Round 1

Reviewer 1 Report

This is an interesting study but a few mechanisms are not clear in its current format which I have listed below;

- Authors stated 'PTT can achieve high accuracy, high efficiency, mild toxicity, and non-invasive treatment compared with traditional chemotherapy, radiotherapy, and surgery' but how? authors should clearly explain this. 

- the light penetration into tissues is a big challenge in light-mediated therapies. Can please authors how they achieve high penetration depth of light into tissues or phantoms? and what is the mechanism of light penetration.

- In introduction, they should also mention the development in light-mediated theranostics and also the use of plasmonic nanoparticles etc, there has been huge development in this area.

- PDT - authors should add that how PDT relies on signlet oxygen and how ROS plus singlet oxygen can address the challenge of hypoxia, they should take help from this paper; 10.1016/j.redox.2017.11.018

- Graphene quantum dots have extensively been used in PDT, authors did not mention this development, please highlight this in one sentence too.

- They mentioned MOF in PDT section but they should add how the attachment of different photosensitisers to MOF has improved the efficacy of PDT. They should list photosensitisers as well.

- Authors mentioned gases, but they should add how NO has been used over the years for treatment of different diseases in clinic and also how the crosstalk between NO and H2S can induce apoptosis and what is the role of light in mediating the controlled, sustained and targeted release of NO.

Author Response

Response to Reviewer#1

Comments and Suggestions for Authors

This is an interesting study but a few mechanisms are not clear in its current format which I have listed below;

Response: We highly appreciate the reviewer for the valuable comments. We have carefully revised our manuscript after reading your constructive advises. The responses to your question are following:

  1. Authors stated 'PTT can achieve high accuracy, high efficiency, mild toxicity, and non-invasive treatment compared with traditional chemotherapy, radiotherapy, and surgery' but how? authors should clearly explain this.

Response: We appreciate the reviewer for the valuable suggestions. As suggested, we have supplied more explanation on our revised manuscript. On our manuscript, we talked about the reasons mainly due to the laser, hyperthermia, as well as photothermal agents.

  1. The light penetration into tissues is a big challenge in light-mediated therapies. Can please authors how they achieve high penetration depth of light into tissues or phantoms? and what is the mechanism of light penetration.

Response: We appreciate the reviewer for the valuable suggestions. As suggested, we introduced the laser penetration depth into tumor tissues on our revised manuscript (NIR-â…  widow laser 1~2 cm, NIR-â…¡ widow laser > 2 cm). In addition, we have given explanation on introduction. The limit penetration depth making it difficult for the tumor site to rise to sufficient temperature is one of reason for choosing mild-temperature photothermal therapy. Thanks.

  1. In introduction, they should also mention the development in light-mediated theranostics and also the use of plasmonic nanoparticles etc, there has been huge development in this area.

Response: We appreciate the reviewer for the valuable suggestions. As suggested, we have introduced the light-mediated theranostics and added the content of plasmonic nanoparticles and references. Thanks.

  1. PDT - authors should add that how PDT relies on signlet oxygen and how ROS plus singlet oxygen can address the challenge of hypoxia, they should take help from this paper; 10.1016/j.redox.2017.11.018

Response: We appreciate the reviewer for the valuable suggestions. As suggested, we have supplied more contents about PDT on our revised manuscript and added the reference. Thanks.

  1. Graphene quantum dots have extensively been used in PDT, authors did not mention this development, please highlight this in one sentence too.

Response: We appreciate the reviewer for the valuable suggestions. As suggested, we have added some sentences about graphene quantum dots on our revised manuscript. Thanks.

  1. They mentioned MOF in PDT section but they should add how the attachment of different photosensitisers to MOF has improved the efficacy of PDT. They should list photosensitisers as well.

Response: We appreciate the reviewer for the valuable suggestions. We explained the PSs as linkers of MOF to reduce the dosage for PDT. For the example at our manuscript, the nanoplatforms combined PDT with PTT to improved the efficacy.

  1. Authors mentioned gases, but they should add how NO has been used over the years for treatment of different diseases in clinic and also how the crosstalk between NO and H2S can induce apoptosis and what is the role of light in mediating the controlled, sustained and targeted release of NO.

Response: We appreciate the reviewer for the valuable suggestions. As suggested, we have explained the strategy about laser mediating the release of NO on our revised manuscript: utilized PAs to combine with photo-triggered NO generator (thiolated transferrin), thus promoting the release of NO under the irradiation of 808 nm near-infrared light. Thanks.

Reviewer 2 Report

The authors submitted a manuscript entitled 'Enhancing the efficiency of mild-temperature photothermal therapy for cancer assisting with various strategies' as a review article in Pharmaceutics. This reviewer found that the mid-temperature PTT is timely and plausible applications for material-based therapeutics. However, the current form requires further polishing to be read to readers.

Content related comments:

-       Section 2 is titled “Various Strategies” but the content/subsection is too broad. In general, this section talks about 1) mechanism of MTPTT and 2) approaches to improve the efficacy of MTPTT. I think it’s better to change the section title to something else, and also separate this part into one part explaining MTPTT in detail, and another explaining the approaches/variables that could be controlled to improve MTPTT.

-       Section 3 is very unbalanced. Subsection 3.1 explained in detail about immunotherapy with nice build-up of explanation in the paragraph, but subsection 3.2 until 3.4 has very limited information and illustration.

-       Reference number 109 was not mentioned anywhere in the manuscript

Image/table related comments:

-       Scheme illustration in Figure 1 is too confusing. What is the meaning of the outer part? Does the difference in color mean anything? What is the correlation between inner part and outer part? Also why is the text upside down text in the bottom part of the scheme?

-       Figure 2 has no caption for part (A).

-       Figure 3 only explains macroautophagy, yet the text that refers to it (Page 5 line 6) talks about types of autophagy

Miscellaneous/proofreading related comments:

-       Wrong grammar usage and typos in manuscript

-       Consistency in text formatting (e.g. Figure 2 in bold characters but Figure 3 isn’t, some sentence written as et. al, some written as et al., etc)

-       Reference writing format need to be fixed

Minor suggestions

(1) Add full words for following abbreviations:

Page 8. Photothermal agent > PA, TME

Page 10. TAT, RGD, near-infrared > NIR

Page 13. BMDC

Page 16. PDT , RT

(2) Please add appropriate references in following sentences:

Page 1. Exogenous photothermal agents (PAs) are not necessary for PTT but can improve the efficiency and efficacy of therapy.

Page 2. Nowadays, a division between the concentration of preclinical and clinical PTT research is obvious, with preclinical studies focused on new PAs, whereas clinical studies concentrated on the exploitation of integrated laser devices.

Page 13. However, immunotherapy is not effective for all tumor types, owing to the toxicity of high immunohorizons and low objective rate.

Page 16. RT can damage DNA and cancer cell death without depth limitation.

Page 17. A high concentration of NO, CO and H2S in the blood can cause poisoning, however, in a relatively mild concentration range, they have significant anti-cancer activity.

Author Response

Response to Reviewer#2

Comments and Suggestions for Authors

The authors submitted a manuscript entitled 'Enhancing the efficiency of mild-temperature photothermal therapy for cancer assisting with various strategies' as a review article in Pharmaceutics. This reviewer found that the mid-temperature PTT is timely and plausible applications for material-based therapeutics. However, the current form requires further polishing to be read to readers.

Response: We highly appreciate the reviewer for the valuable comments. We have carefully revised our manuscript after reading your constructive advises. The responses to your question are following:

  1. Content related comments:

(1) Section 2 is titled “Various Strategies” but the content/subsection is too broad. In general, this section talks about 1) mechanism of MTPTT and 2) approaches to improve the efficacy of MTPTT. I think it’s better to change the section title to something else, and also separate this part into one part explaining MTPTT in detail, and another explaining the approaches/variables that could be controlled to improve MTPTT.

Response: We appreciate the reviewer for the valuable suggestions. As suggested, we have changed it on our revised manuscript. Thanks.

(2) Section 3 is very unbalanced. Subsection 3.1 explained in detail about immunotherapy with nice build-up of explanation in the paragraph, but subsection 3.2 until 3.4 has very limited information and illustration.

Response: We appreciate the reviewer for the valuable suggestions. As suggested, we have changed the sequence of section 3 on our revised manuscript. Thanks.

(3) Reference number 109 was not mentioned anywhere in the manuscript

Response: We appreciate the reviewer for the valuable suggestions. As suggested, we have now corrected the reference. Thanks.

  1. Image/table related comments:

(1) Scheme illustration in Figure 1 is too confusing. What is the meaning of the outer part? Does the difference in color mean anything? What is the correlation between inner part and outer part? Also why is the text upside down text in the bottom part of the scheme?

Response: We appreciate the reviewer for the valuable suggestions. As suggested, we have now corrected Figure 1. Thanks.

(2) Figure 2 has no caption for part (A).

Response: We appreciate the reviewer for the valuable suggestions. As suggested, we have now added it.

(3) Figure 3 only explains macroautophagy, yet the text that refers to it (Page 5 line 6) talks about types of autophagy

Response: We appreciate the reviewer for the valuable suggestions. We mainly introduced the relationship between macroautophagy with MTPP because we only found that macroautophagy could augment the efficacy of MTPTT from references. In addition, on the section 2.5 autophagy mediation, we carefully explained how autophagy to impact MTPTT for cancer. So we thanks for your suggestion.

  1. Miscellaneous/proofreading related comments:

(1) Wrong grammar usage and typos in manuscript

Response: We appreciate the reviewer for the valuable suggestions. As suggested, we have now checked and revised the manuscript. Thanks.

(2) Consistency in text formatting (e.g. Figure 2 in bold characters but Figure 3 isn’t, some sentence written as et. al, some written as et al., etc)

Response: We appreciate the reviewer for the valuable suggestions. As suggested, we have now corrected it.

(3) Reference writing format need to be fixed

Response: We appreciate the reviewer for the valuable suggestions. As suggested, we have now corrected the reference styling of the revised manuscript following the Pharmaceutics guidelines.

  1. Minor suggestions:

(1) Add full words for following abbreviations:

Page 8. Photothermal agent > PA, TME

Page 10. TAT, RGD, near-infrared > NIR

Page 13. BMDC

Page 16. PDT , RT

Response: We appreciate the reviewer for the valuable suggestions. As suggested, we have now added it.

(2) Please add appropriate references in following sentences:

Page 1. Exogenous photothermal agents (PAs) are not necessary for PTT but can improve the efficiency and efficacy of therapy.

Page 2. Nowadays, a division between the concentration of preclinical and clinical PTT research is obvious, with preclinical studies focused on new PAs, whereas clinical studies concentrated on the exploitation of integrated laser devices.

Page 13. However, immunotherapy is not effective for all tumor types, owing to the toxicity of high immunohorizons and low objective rate.

Page 16. RT can damage DNA and cancer cell death without depth limitation.

Page 17. A high concentration of NO, CO and H2S in the blood can cause poisoning, however, in a relatively mild concentration range, they have significant anti-cancer activity.

Response: We appreciate the reviewer for the valuable suggestions. As suggested, we have now added it. Thanks.

Round 2

Reviewer 1 Report

Authors have elegantly addressed my concerns and I am pleased to recommend the revised submission for publication in Pharmaceutics.